# Prediction of Crack Formation for Cross Wedge Rolling of Harrow Tooth Preform

**DOI:** 10.3390/ma12142287

**Published:** 2019-07-17

**Authors:** Zbigniew Pater, Janusz Tomczak, Tomasz Bulzak, Jarosław Bartnicki, Arkadiusz Tofil

**Affiliations:** Department of Computer Modelling and Metal Forming Technologies, Faculty of Mechanical Engineering, Lublin University of Technology, Nadbystrzycka 36, 20-618 Lublin, Poland

**Keywords:** cross wedge rolling, ductile fracture, FEM, experiment

## Abstract

The article presents the issue of material fracture during the process of cross-wedge rolling (CWR). The object of the research was the process of forming a harrow tooth preform. In the conducted analysis nine damage criteria were applied. The critical value of damage was determined with a new calibrating test, basing on rotational compression of a sample in a channel. The results of calculations were compared to the results of experimental testing performed in laboratory conditions in Lublin University of Technology. On the basis of the obtained results an assessment of the applied damage criteria and their applicability in the analysis of CWR processes was conducted.

## 1. Introduction

Cross-wedge rolling (CWR) is an innovative manufacturing technology, applied in the manufacturing of stepped shafts and axles as well as preforms for die forging. The CWR process offers numerous advantages, amongst others: High efficiency, low material waste, accurateness of the obtained products, easy automation and eco-friendliness [1,2].

Cross-wedge rolling has been used in the industry for c.a. 50 years and since then has been developed in terms of theory and manufacturing technology. The majority of the developed aspects [3,4,5] were connected to the hindrances in the forming process. Among those hindrances there are: Uncontrolled slipping, necking (rupture) of the rolled step and material cracking in the axial area of the workpiece. It is also to be mentioned that the most significant problems are connected to estimating the occurrence of the last hindrance, which is cracking.

The first works on this subject were published in the beginning of the 21st century. In 2002 Li et al. in their work [6] introduced the results of an analysis stating that the phenomenon of microfractures melding into macrofractures is caused by the occurrence of shear and tensile stresses, which cause the cracking to be cross-shaped. Two years later Li and Lovell [7] determined the influence of the basic cold CWR parameters on the occurrence of cracks, using the same numerical model. According to the obtained results material cracking was influenced by increasing the wedge angle (forming *α* and spreading angle *β*) and increasing the reduction of the cross section *R_p_* (degree of reduction δ).

Later on, numerical modeling became applied in analyses of hot CWR processes. In 2005 Piedrahita et al. [8] conducted an analysis of several cases of rolling using FORGE® software in order to assess the influence of basic CWR parameters on the Cockroft–Latham damage function. As a result of the research it was stated that material cracking during CWR is most likely to occur when wedges with more significant *α* and *β* angles are employed as well as when less significant cross section reductions are used. Similar conclusions were drawn by Pater in his work [1] published in 2014. According to Yang et al. [9] both significant (*R_p_* > 75%) and insignificant (*R_p_* < 30%) reductions of the cross-section facilitate the phenomenon of material cracking in CWR. The cross-sectional reduction *Rp* was calculated with the following formulas:(1)$$R_{p}=\left(1-\frac{d^{2}}{d_{0}^{2}}\right)\cdot 100\%,$$
where *R_p_*—reductions of the cross-section during CWR, *d_0_*—diameter of the billet and *d*—diameter of the rolled step.

Lately numerous scientific articles presenting the distributions of damage function in CWR-formed elements were published. In those articles, the following criteria were employed: Oyane–Sato [10], Oh [11], Johnson–Cook [12,13] and Cockroft–Latham [8,14,15,16,17]. In order to determine whether a fracture will occur in a given forming case both the distribution of the damage function and the critical damage value, obtained in so-called calibrating tests, are required. The currently employed tests [18,19], based on tension, compression and torsion cannot be employed in a CWR analysis due to significant differences in between the stress state of the test and the real process. For this reason a new test, based on rotational compression was developed in Lublin University of Technology. In the aforementioned test material cracking occurs in the axial area of the sample due to the occurrence of the Mannesmann effect [20]. Upon obtaining the critical damage values one may prognosticate the cracking of the material in the CWR process.

This article presents the results of a numerical analysis aiming to determine the areas in which cracking occurs during the rolling process of a harrow tooth preform. In the research nine damage criteria were used. Their critical damage values were determined using rotational compression in a channel.

## 2. Subject of Research 

The subject of the analysis was the process of manufacturing a harrow tooth preform presented in Figure 1 along with a forging. The preform is obtained by flashless forging in a hydraulic press. During forging the working part of the tooth (with a square cross-section) as well as a flange with a cam is formed and finished. The gripping part, securing the tooth, is machined in the next operations.

Upon analyzing the shape of the preform it was to be noticed that it’s manufacturing using the CWR method might pose difficulties. Assuming that forming will be performed from a cylindrical billet with a 22 mm diameter it is to be mentioned that the forming process of the cone would occur at a very high values of strain (maximum value of reduction ratio is equal δ = 5.5, where δ = *d*_0_/*d*; *d*_0_—diameter of the billet and *d*—diameter of the rolled step). The flange, on the other hand, must be formed by rolling with upsetting, which allows one to obtain a diameter bigger than the one of the billet. An additional hindrance to the process is a relatively significant length of the preform, which is rolled on its whole length. Due to this fact, relatively big spreading angle *β* must be employed.

Preform rolling was conducted in a flat-wedge rolling mill located in Lublin University of Technology, shown in Figure 2. The rolling mill has a hydraulic drive and is equipped with a set of one (upper) movable tool, working at the speed *v* = 300 mm/s. Maximum length of the tools (wedges) is equal 1 m.

Figure 3 presents one of the wedge tools securing the preform rolling process. The forming wedge is characterized by the angles *α* = 20° and *β* = 9.5°. The side surfaces of the wedges have technological cuts, aiming to hinder the occurrence of uncontrollable slipping. In the ends of those tools two knives for cutting end waste are located.

In the process of manufacturing preforms billets with the dimensions Ø22 × 140 mm, preheated in an electric chamber furnace to 1150 °C were used. The billets were placed on the nether tool. Upon activating the movement of the slide of the rolling mill, the upper wedge came in contact with the workpiece, which was rolled on the stationary tool. In the final stage of the rolling process the side waste was removed and the preform was obtained. The progress of the discussed CWR process is shown in Figure 4.

The preform manufactured using CWR is shown in Figure 5. Its shape was compliant with the assumptions and its dimensions remain in the range of tolerance ±0.5 mm. The preform was subjected to non-destructive testing for internal cracking, conducted in an X-ray machine X-VIEW model X25 (NSI North Star Imaging Inc., Rogers, USA). The tests revealed cracking in the griping part of the preform (Figure 6). The observed cracks were longitudinal, c.a. 24 mm long and began c.a. 4 mm after the step, from which the cam was formed in the forging process. In the cross-section the cracks were of an irregular, dense shape and their size did not exceed 3 mm. No cracks were observed in the remaining part of the preform (also in its conical end, presented in Figure 7).

The detected cracks did not disqualify this method of forming the harrow tooth, implemented in one of the Polish industry plants. Due to their occurrence in the axial area as well as the type of load during its operation (bending) the lifetime of the teeth was not influenced. In this article the cracks were used determine the possibility of prognosticating the material fracture using numerical modeling.

## 3. Modeling of the Ductile Fracture

According to the results published by Yang et al. [21] ductile fracture may occur in elements formed by the CWR method. The most popular method of modeling ductile fracture is, in this case, monitoring the damage function *f_i_* described by the following general dependency:(2)$$f_{i}=\int_{0}^{\varepsilon_{f}}\Phi {\left(\sigma \right)}_{i}d\varepsilon ,$$
where $\Phi {\left(\sigma \right)}_{i}$—function describing the influence of the strain state on fracture in *i* criterion, and *ε_f_*—critical strain.

In order to prognosticate material cracking, value of critical damage *C_i_* of the damage function *f_i_* at the moment of cracking is required. With the present value of the function *f_i_* and the critical value *C_i_* fracture indicator *w_i_* can be determined. This indicator shows the degree of exposure to cracking expressed in percent. This indicator is determined from the following dependency: (3)$$w_{i}=100\%\frac{f_{i}}{C_{i}}.$$
For the crack to occur $w_{i}\geq 100\%$.

In the specialist literature numerous damage criteria varying in the used function $\Phi {\left(\sigma \right)}_{i}$ can be found. A list of those functions, presented in chronological order is shown in Table 1, along with a method of designating for each criterion. The aforementioned damage criteria were applied further in this study.

In order to successfully model fracture the critical damage value *C_i_*, obtained in calibrating tests must be known. In the case of the CWR process it is advised to use the rotational compression in a channel test, a scheme of which is presented in Figure 8. In this test a disc-shaped sample with the dimensions *d*_0_ × *b* is placed in a channel of the nether, unmovable tool. The upper tool is then set in reciprocating motion and grips the sample (due to the cut of the *γ* angle) and compresses it in the radial direction. Compression is possible due to the fact that the distance between the bottoms of the channels of both tools (marked as 2*h*) is smaller than the diameter of the disc *d*_0_.

As a result of the friction forces the sample is rolled in the channel of the nether tool on the length of the path *s* and is subjected to the cyclic compressive-tensile stress [20]. During forming the side walls of the channels hinder material flow to the sides and therefore ensures continuous compression of the sample on the entire length of the forming path. The test aimed to experimentally determine the length of the *s* path, at which the crack occurs for the assumed value of the degree of reduction *δ*, in this case described as:(4)$$\delta =\frac{d_{0}}{2h}.$$

Then the performer test ought to be simulated using FEM and the value of the damage function in the axis of the sample ought to be determined. This value is equal to the sought critical damage value.

Figure 9 presents one of the rotational compression tests, performed in order to determine the critical damage values (for the nine discussed criteria, shown in Table 1) for C45 grade steel formed in 1150 °C. The length of the forming path in the test case was equal *s* = 700 mm. The samples of the dimensions Ø40 × 20 mm were heated in an electric resistance furnace and compressed at *δ* = 1.053. The speed of the movable jaw was 300 mm/s. It is also to be mentioned that despite a relatively long forming time (*t* = 4.7 s) the material in the middle of the sample did not cool down. Heat was transferred to the tools only from the peripheral part of the sample, whereas in the axial area a substantial amount of heat was created as a result of the exchange of the work of plastic strain.

The research allowed the authors to determine that the sought length of the path s at which the material cracking occurs is equal 700 mm (Figure 10). For this path a simulation in Simufact.Forming v.15 software was conducted.

In the calculations the material model of the formed C45 grade steel is described by the following equation [32]:(5)$$\sigma_{F}=2859.85e^{-0.003125T}\varepsilon^{\left(0.00004466T-0.10126\right)}e^{\left(-0.00002725T+0.0008183\right)/\varepsilon }{\dot{\varepsilon }}^{\left(0.00015115T-0.002748\right)},$$
where *σ_F_*—flow stress, MPa; *ε*—effective strain, -; $\dot{\varepsilon }$—strain ratio, s^−1^ and *T*—temperature, °C.

Moreover, the Tresca friction model, determined by friction factor *m* = 0.8 was assumed. Considering the thermal phenomena occurring during the forming process it was presupposed that during the process the temperature of tools is constant and equal to 50 °C, whereas the heat transfer coefficient between the tools and the material is equal to 10000 W/m^2^K. The remaining parameters were set similarly to the experimental testing.

Figure 11 shows the process of the numerically simulated rotational compression in a channel. In order to enhance the demonstrability, the upper tool was hidden. Additionally the change to temperature of the material in the sample during forming was presented. Interestingly, in the axial area of the sample material temperature not only did it not decrease, but also increased by over 10 °C. Contrastingly, the temperature of the outer (peripheral) surfaces of the disc decreased by c.a. 100–150 °C.

In order to determine the value of the damage function 11 virtual sensors were placed in the axis of the sample, located initially every 2 mm. This way the parameters determining the state of stress and strain were measured. As a result of forming the front surfaces of the sample funnels appeared, as a result of which the thickness of the sample in the axial area was decreased (Figure 12). This type of strain was deemed beneficial for the test, since the cracking was occurring very fast on the side surface of the sample and observation of the sample was enough to determine its occurrence.

The data recorded using the sensors allowed one to determine the distributions of the damage function in the axis of the sample, after it being subjected to rotational compression on the path *s* = 700 mm. In order to achieve this, the fracture criteria presented in Table 1 were used. The obtained distributions *f_i_* are shown in Figure 13 and Figure 14. In order to obtain the critical values of damage, the obtained distributions of the damage function were averaged and the results were listed in Table 2.

## 4. Numerical Modeling of the Process of Cross Wedge Rolling of the Harrow Tooth Preform

Upon establishing the critical damage values for the nine *f_i_* functions the authors conducted the numerical simulation of the CWR process of the harrow tooth preform. The simulation was performer in Simufact.Forming v.15, a software that has been used by the authors to model cross and skew rolling processes [33,34,35,36,37]. The obtained results remained in compliance with the results of the verifying experimental examination.

Figure 15 presents a geometrical model of the analyzed case of rolling, which accurately portrayed the tools and the billet used in the forming tests performer in laboratory conditions in Lublin University of Technology. Both tools were assumed to be perfectly rigid bodies, whereas the billet modeled by 8-node brick elements behaved as a rigid-elastic material. The material model of the C45 grade steel, described by dependency (5) was identical to the one applied in determining the critical damage values. Similarly to this, the friction factor was also equal *m* = 0.8 in this case. A comparison of the remaining parameters assumed in this simulation comprises of a billet temperature 1150 °C, temperature of the tools 50 °C, displacement rate of the movable tool 300 mm/s, heat transfer coefficient between the material and the tools 10,000 W/m^2^K.

The conducted numerical simulation allowed for an accurate remodeling of the preform rolling process, as shown in Figure 16. During the process an insignificant amount of side waste was created and removed in the final stage of the rolling process. Moreover, Figure 16 shows the change to effective strain on the surface of the workpiece. Upon analyzing the strain values it was observed that it reached the highest values in the conical part—the area of the most significant reduction of the cross section.

The relatively high values of effective strain are located in the areas of the preform close to the flange, where significant material flow in the tangential (peripheral) direction occurred.

The aim of the simulation was to assess the efficiency of the modeling of cracking in the CWR process. In the calculations the fracture criteria shown in Table 1 were used. Due to the fact that only a few of the criteria are implemented in Simufact.Forming, a similar methodology to the one adopted in the process of determining the critical values for rotational compression was assumed. In the axis of the rolled workpiece 68 virtual sensors were placed to monitor the components of the stress and strain state, necessary to determine the values of each damage function *f_i_*. The way in which the sensors were located both in the billet and in the rolled element are shown in Figure 17.

Every damage function used in the calculations is related to energy, which is a result of the integration of the stress after strain. Therefore an examination of the distribution of those parameters in the rolled preform was deemed important. The distribution of effective strain, shown in Figure 18 proved rather uncomplicated. The obtained data shows that the least significant strain occurs in the flange, where the material is being upset. A slightly higher strain value was observed in the near-flange area, from where the material was extruded into the flange. A relatively stable distribution of strain occurred in the cylindrical steps of the product. A very significant increase of strain, with the highest values (ε=13.5) occurred in the end of the cone.

Illustrating the state of strain in the axis of the rolled preform proved more challenging. The chosen method was a presentation of the distributions of stress triaxiality *η* and Lode angle parameter *θ*, connected to the invariants of the state of stress, which play a significant role in the material cracking [38,39,40]. Stress triaxiality *η* is described by the ratio of the first invariant *p* to the second invariant *q* and is expressed by the following equation:(6)$$\eta =\frac{-p}{q}=\frac{\sigma_{m}}{\sigma_{i}},$$
where *σ_m_*—mean stress and *σ_i_*—equivalent stress.

The Lode angle parameter, however, is represented by the second *p* and third *r* invariants of the state of strain *r*, and is expressed by the following dependency:(7)$$\theta =1-\frac{2}{\pi }\arccos\left[{\left(\frac{r}{q}\right)}^{3}\right],$$
where:(8)$$r={\left[\frac{27}{2}\left(\sigma_{1}-\sigma_{m}\right)\left(\sigma_{2}-\sigma_{m}\right)\left(\sigma_{3}-\sigma_{m}\right)\right]}^{\frac{1}{3}}.$$
In the above presented dependency σ_1_, σ_2_ and σ_3_ express principal stresses.

Since the values of both *η* and *θ* change significantly during the forming process, average values of those parameters were determined in each virtual sensor. In order to achieve this, the following dependencies were applied:(9)$$\eta_{av}=\frac{1}{\varepsilon }\int_{0}^{\varepsilon }\eta \;d\varepsilon ,$$
(10)$$\theta_{av}=\frac{1}{\varepsilon }\int_{0}^{\varepsilon }\theta \;d\varepsilon .$$

The distributions *η_av_* and *θ_av_*, obtained on the basis of the above presented dependencies, are shown in Figure 19. The analysis of changes to stress triaxiality indicates that on the majority of the preform’s length (apart from the cone) $\eta_{av}\geq 0.33$. According to the test results [41,42] in the case of the state of stress compliant with this condition material cracking was caused by void formation. In the case when $0<{\;\eta }_{av}<0.33$ the cracks may occur as a result of both void formation and shear fracture. Such a situation occurs during the forming of the working part of the preform (cone), where η_av_ decreased to 0.172 along with the increase in the reduction of cross section. As far as the Lode angle parameter *θ_av_* is concerned, it was observed that its value varied in the range –0.539 to 0.015. In the vast majority of the preform the values of *θ_av_* were negative, the smallest in the area where the forming process began. Such values of the Lode angle parameter indicate the occurrence of an intermediate state between the axisymmetric compression, where *θ* = –1, and the generalized shear, where *θ* = 0.

Obtaining the components of the stress and strain state allowed one to determine the final values of each damage function *f_i_* in the virtual sensors. Then, the value of critical damage indicator *w_i_* was calculated by dividing the obtained *f_i_* damage function by the critical damage values *C_i_*. Lastly, after considering the changes to the location of each sensor, the distributions of the *w_i_* indicators in the axis of the preform were determined and are shown in Figure 20. In the areas where $w_{i}\geq 100\%$ a fracture was prognosticated to occur. 

In order to simplify the comparison of the results of modeling with the experimental testing results an additional Figure 21 was made, where the areas of crack occurrence, prognosticated using the nine criteria, were shown. An analysis of the data presented in this figure showed that all of the applied criteria erroneously indicated cracking in the working part (cone) of the preform, that is in the areas where stress triaxiality $\eta_{av}<0.33$. Assuming energetic models of fracture in this case led, due to very significant values of effective strain, to a major overestimation of the damage function. It seems that in such state of stress criteria ought to be applied, e.g., maximum shear stress criterion.

A contrary situation occurs in the case of prognosticating fracture in the remaining part of the preform, where $\eta_{av}\geq 0.33$. In this case applying four out of nine allowed us to indicate the cracking in the proper pace of the gripping part of the preform. One of those criteria (developed by Ayada) falsely indicated that cracking would occur in the cylindrical part of the preform. It was concluded that the accurate indication of fracture occurrence in the gripping part of the preform was developed by: Freudenthal, Cockroft and Latham and Argon et al. The efficiency of those criteria is limited to the state of stress characterized by $\eta_{av}\geq 0.33$, which is the cases in which fracture is caused by void formation.

## 5. Conclusions

Based on the research, the following conclusions were drawn:Material cracking in the axial area of the workpiece was one of the major hindrances to the CWR process;In order to prognosticate the material cracking the critical damage of the material must be known. In the case of the CWR process it could be determined based on the rotational compression of a disc in a channel; in this article this method was described in detail on the example of determining the critical damage values for C45 grade steel, formed in 1150 °C;In prognosticating material fracture during the CWR process realized in the state of stress characterized by stress triaxiality $\eta_{av}\geq 0.33$, the widely-known damage energetic criteria could be applied; the best results were obtained using the criteria developed by Freudenthal, Cockroft and Latham and Argon et al.;In the case of CWR realized in the stress triaxiality $\eta_{av}<0.33$ applying energetic damage criteria proved unsuccessful; it is to be deemed reasonable to seek for damage criteria of a different type (e.g., stress), whose efficiency ought to be researched in other papers.

## Figures and Tables

**Figure 1 materials-12-02287-f001:**
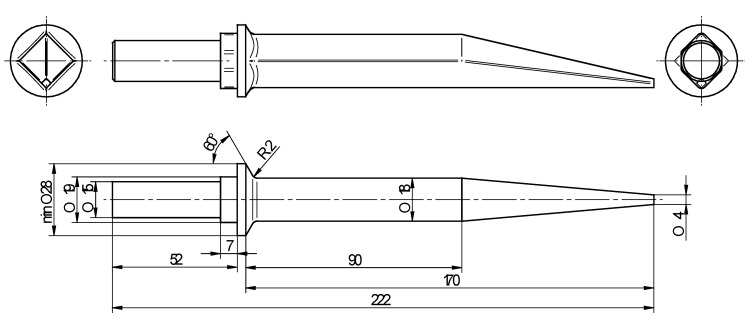
Harrow tooth forging (on top) and preform manufactured by cross-wedge rolling (on the bottom).

**Figure 2 materials-12-02287-f002:**
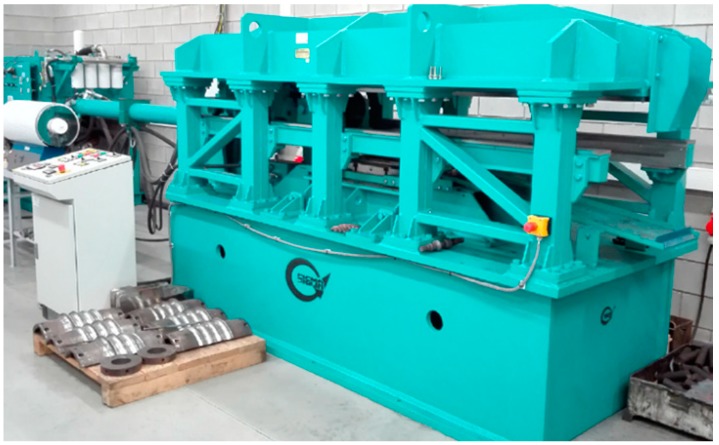
Cross-wedge rolling mill used in experimental testing.

**Figure 3 materials-12-02287-f003:**
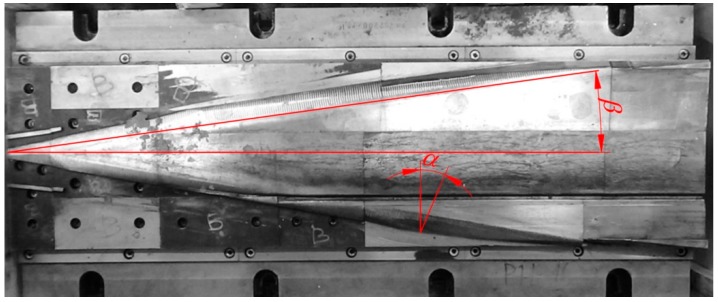
One of the wedges used in the test rolling of the harrow tooth preform.

**Figure 4 materials-12-02287-f004:**
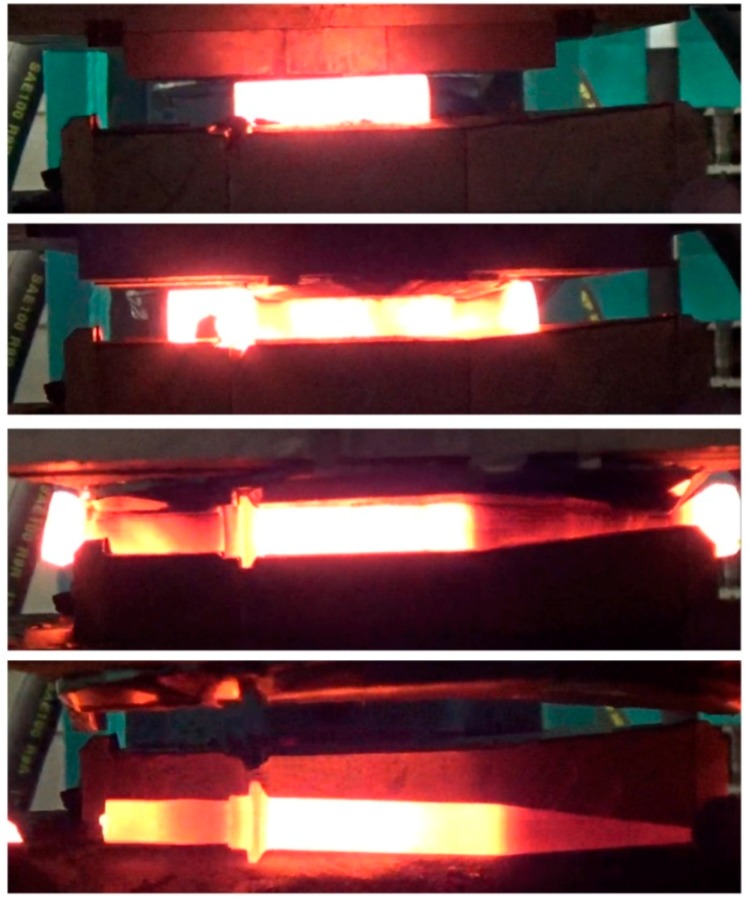
The cross-wedge rolling (CWR) process of the harrow tooth preform conducted in laboratory conditions in the Lublin University of Technology.

**Figure 5 materials-12-02287-f005:**
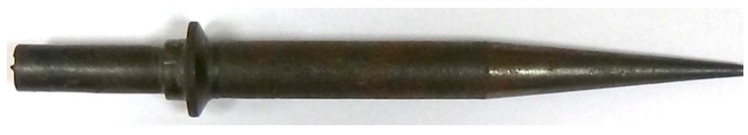
Preform of the harrow tooth obtained in the researched CWR process.

**Figure 6 materials-12-02287-f006:**
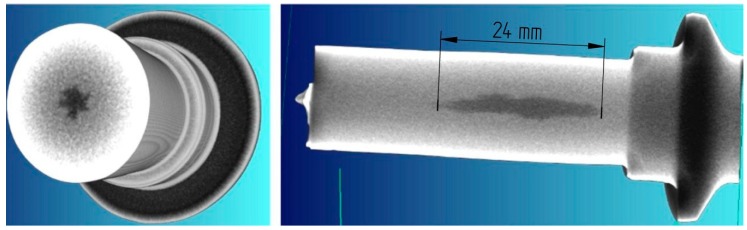
Results of the CT scan of the gripping part of the harrow tooth preform.

**Figure 7 materials-12-02287-f007:**
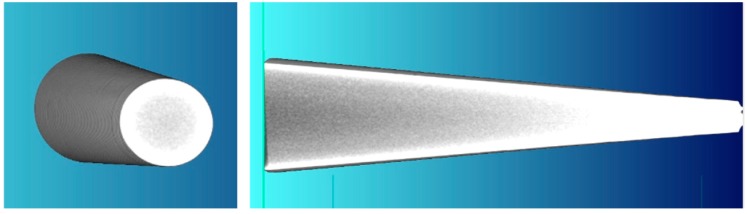
Results of the CT scan of the working (conical) part of the harrow tooth preform.

**Figure 8 materials-12-02287-f008:**
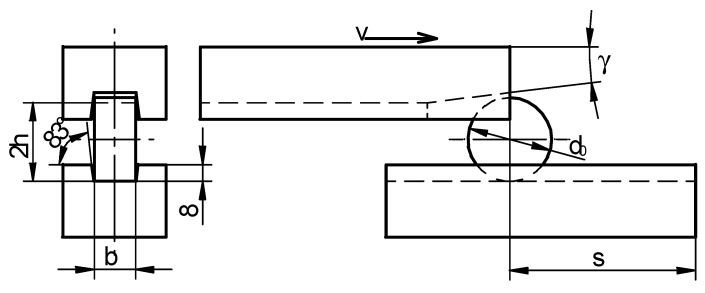
Scheme of the calibrating test of the damage function conducted using the method of rotational compression in a channel.

**Figure 9 materials-12-02287-f009:**
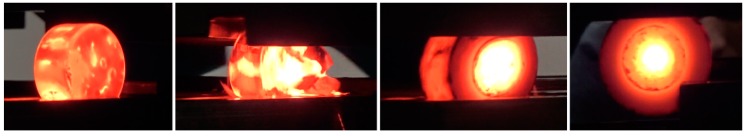
Process of the rotational compression in a channel (path *s* = 700 mm) of a sample heated to 1150 °C, where (respectively from the left): Sample location; gripping the sample by the movable jaw and flaking of a scale; rotational compression of the sample; ejecting the sample from the working area of the tools.

**Figure 10 materials-12-02287-f010:**
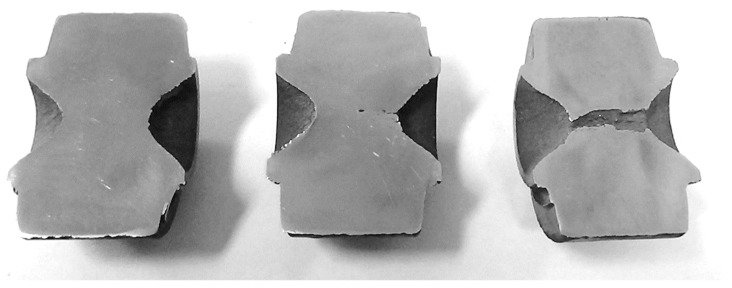
Samples heated to 1150 °C and compressed rotationally in a channel on the path *s* equal (respectively from the left): 700 mm, 750 mm and 800 mm.

**Figure 11 materials-12-02287-f011:**
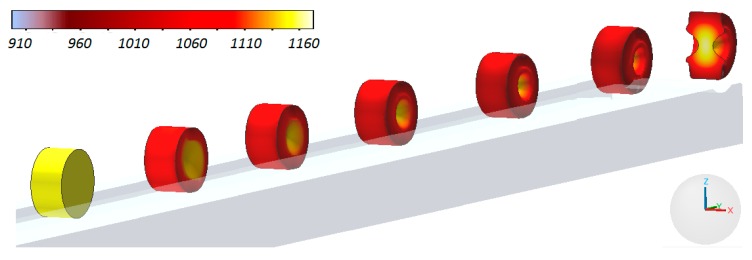
Progression of the sample shape during rotational compression in a channel on the path *s* = 700 mm with the material temperature distribution.

**Figure 12 materials-12-02287-f012:**
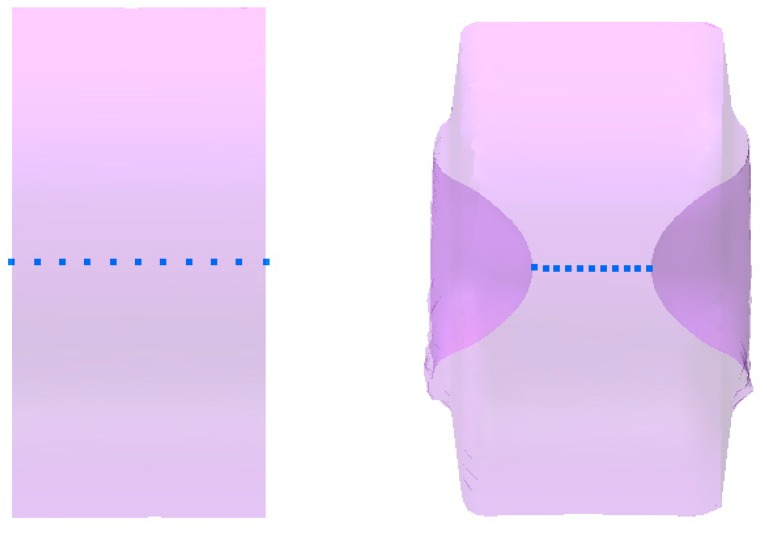
Virtual sensors located in the axis of the sample (every 2 mm) used to record the data characterizing the state of stress and strain: On the left—initial sample and right—the sample after rotational compression.

**Figure 13 materials-12-02287-f013:**
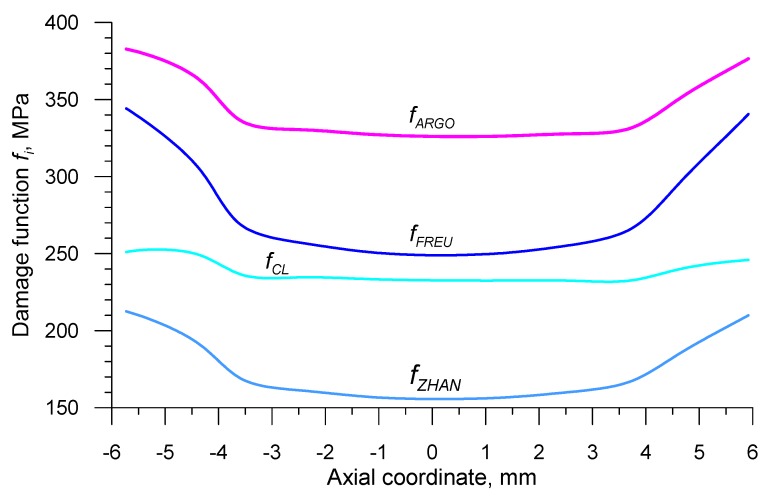
Distributions of the stress damage functions in the axis of the sample subjected to rotational compression in a channel at *s* = 700 mm and *T* = 1150 °C.

**Figure 14 materials-12-02287-f014:**
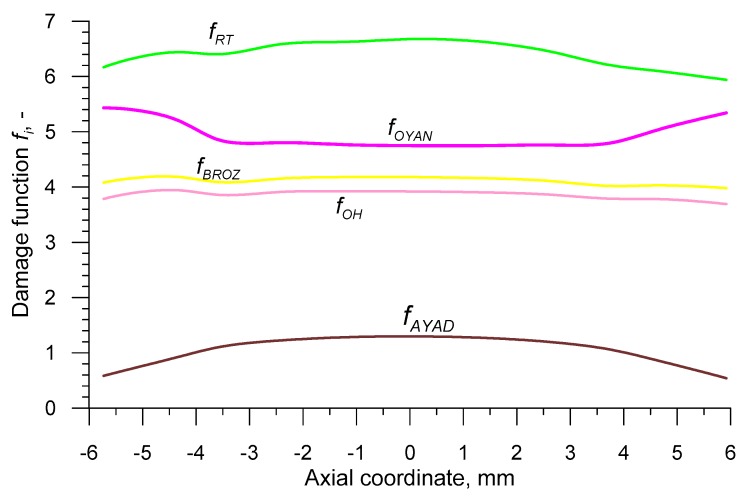
Distributions of the non-dimensional damage functions in the axis of the sample subjected to rotational compression in a channel at *s* = 700 mm and *T* = 1150 °C.

**Figure 15 materials-12-02287-f015:**
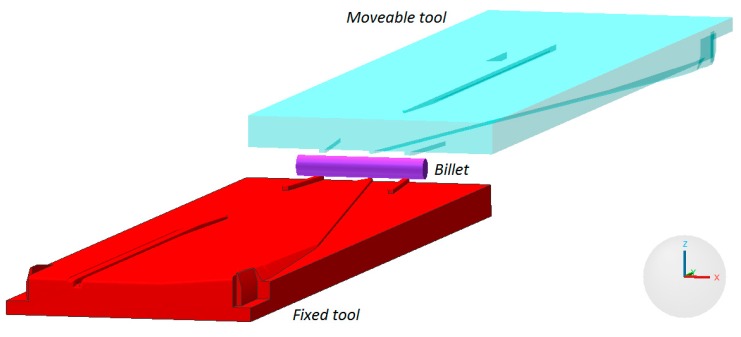
Geometrical model of the CWR process of the harrow tooth preform, realized from the billet with 22 mm diameter.

**Figure 16 materials-12-02287-f016:**
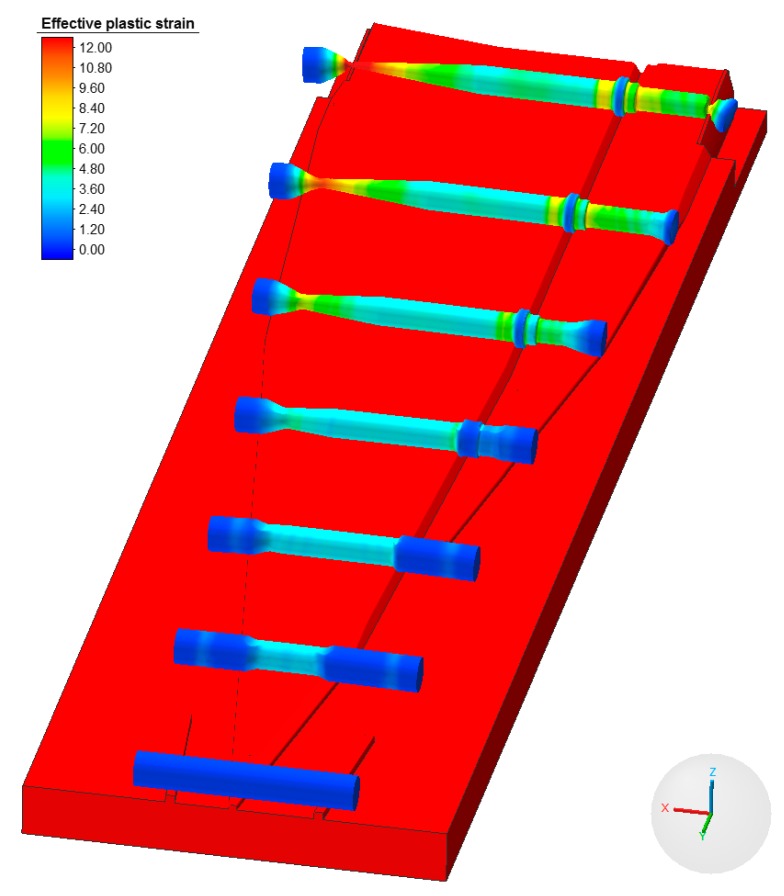
Progression of the shape of the harrow tooth preform during the rolling process with the distribution of effective strain.

**Figure 17 materials-12-02287-f017:**
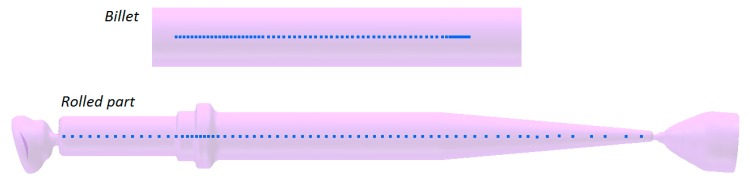
Location of the virtual sensors in the billet and the rolled part, used for monitoring the state of stress and strain.

**Figure 18 materials-12-02287-f018:**
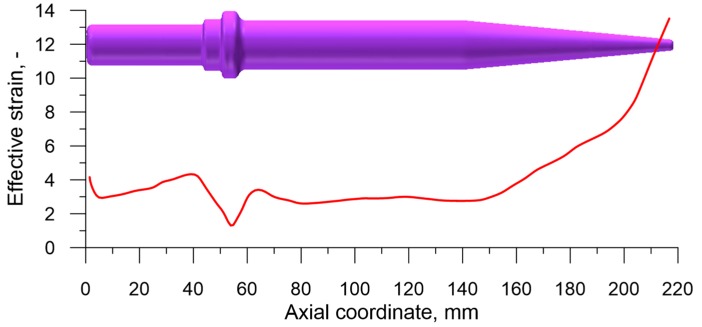
Distribution of the effective strain in the axis of the rolled preform.

**Figure 19 materials-12-02287-f019:**
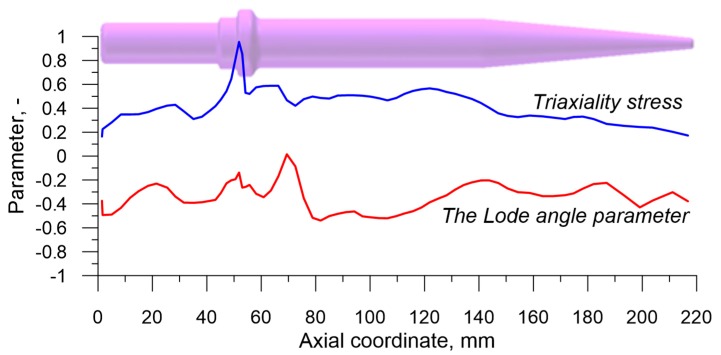
Characteristics of the stress state in the axis of the rolled part expressed by the averaged values of stress triaxiality and the Lode angle parameter.

**Figure 20 materials-12-02287-f020:**
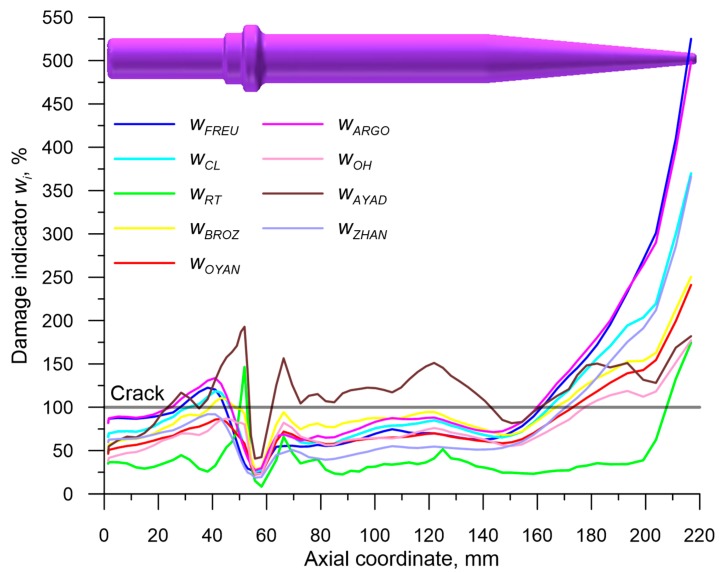
Distributions of the damage function in the axis of the analyzed preform, manufactured by the CWR method.

**Figure 21 materials-12-02287-f021:**
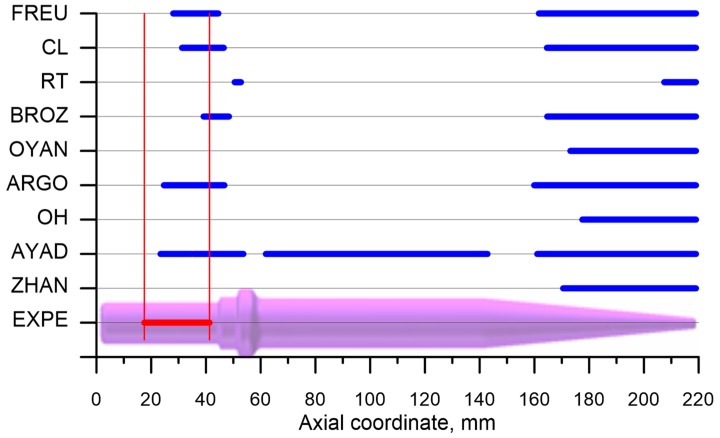
Comparison of the lengths of the experimentally determined cracks (EXPE) and numerically prognosticated ones based on the discussed criteria (marks according to Table 1).

**Table 1 materials-12-02287-t001:** Selected criteria of ductile fracture used for the presented analysis [22,23,24,25,26,27,28,29,30].

Abbreviation	Criterion (year)	Formula
FREU	Freudenthal (1950)	$f_{FREU}=\int_{0}^{\varepsilon }\sigma_{i}d\varepsilon$
CL	Cockroft and Latham (1968)	$f_{CL}=\int_{0}^{\varepsilon }\sigma_{1}d\varepsilon$
RT	Rice and Tracey (1969)	$f_{RT}=\int_{0}^{\varepsilon }\exp\left(\frac{3}{2}\eta \right)d\varepsilon$
BROZ	Brozzo et al. (1972)	$f_{BROZ}=\int_{0}^{\varepsilon }\frac{2\sigma_{1}}{3\left(\sigma_{i}-\sigma_{m}\right)}d\varepsilon$
OYAN	Oyane (1972)	$f_{OYAN}=\int_{0}^{\varepsilon }\left(1+A\eta \right)d\varepsilon$
ARGO	Argon et al. (1975)	$f_{ARGO}=\int_{0}^{\varepsilon }\left(\sigma_{m}+\sigma_{i}\right)d\varepsilon$
OH	Oh et al. (1979)	$f_{OH}=\int_{0}^{\varepsilon }\frac{\sigma_{1}}{\sigma_{i}}d\varepsilon$
AYAD	Ayada (1984)	$f_{AYAD}=\int_{0}^{\varepsilon }\eta d\varepsilon$
ZHAN	Zhan et al. (2009)	$f_{ZHAN}=\int_{0}^{\varepsilon }\left(\sigma_{i}-\sigma_{m}\right)d\varepsilon$

where *A*—material constant (after [31] *A* = 0.424), *σ_m_*—mean stress, *σ_i_*—equivalent stress, *σ*_1_ —maximal principal stress and *η*—stress triaxiality.

**Table 2 materials-12-02287-t002:** Critical damage values for C45 grade steel formed in 1150 °C, determined based on rotational compression in a channel.

*C_FREU_*	*C_CL_*	*C_RT_*	*C_BROZ_*	*C_OYAN_*	*C_ARGO_*	*C_OH_*	*C_AYAD_*	*C_ZHAN_*
180.1 MPa	238.2 MPa	6.38	4.11	4.95	343.3 MPa	3.85	1.03	174.9 MPa

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
