# Peer review of "Prediction of Crack Formation for Cross Wedge Rolling of Harrow Tooth Preform"

_materials, 2019, doi:10.3390/ma12142287_

Reviewer 1 Report

-    The term "damage" I would replace by term “the creation of micro cracks and cracks” (discontinuities) in CWR.

-   The critical damage values to replace by the critical strain values, that generate the cracks.

-    The heating process in furnace at 1150 ° C may produce oxides at the surface. Was the inert atmosphere in the furnace used for heating?

 -   In conclusion, it would be useful to compare the current technology of forging with CWR.

 -    In my opinion, the process of creation crack is most affected by the friction between the tool and the workpiece and by the temperature change in the formed material. These aspects of the process have not been specified by the authors.

Author Response

Thank you for the review of the article and good comments.

Reviewer 1

-    The term "damage" I would replace by term “the creation of micro cracks and cracks” (discontinuities) in CWR.

Response: The term "damage" is commonly used in cross-wedge rolling and other metal forming processes. Therefore, I would not introduce new terminology. Below are a few literature references where term damage is used.

1. M. L. N. Silva, G. H. Pires, S.T. Button. Damage evolution during cross wedge rolling of steel DIN 38MnSiVS5. Procedia Engineering 10 (2011) 752–757.

2. Novella, M.F.; Ghiotti, A.; Bruschi, S.; Bariani, P.F. Ductile damage modeling at elevated temperature applied to the cross wedge rolling of AA6082-T6 bars. J. Mater. Process. Techn. 2015, 222, 259-267.

3.  Huang, X.; Wang, B.; Zhou, J.; Ji, H.; Mu, Y.; Li, J. Comparative study of warm and hot cross-wedge rolling: numerical simulation and experimental trial. Int. J. Adv. Manuf. Technol. 2017, 92, 3541-3551.

4. Yang, C.; Dong, H.; Hu, Z. Micro-mechanism of central damage formation during cross             wedge rolling. J. Mater. Process. Techn. 2018, 252, 322-332.

- The critical damage values to replace by the critical strain values, that generate the cracks.

Response: The critical value of damage is not only a function of the strain state but also of the stress state. This is confirmed by the formulas contained in Table 1. Therefore, the replacement of the critical value of the damage function with the value of strain will not be adequate.

- The heating process in furnace at 1150 ° C may produce oxides at the surface. Was the inert atmosphere in the furnace used for heating?

Response: The process of heating steel to 1150 °C causes oxidation the surface. In the heating process, no protective atmosphere was used. In industrial conditions for heating steel for hot forging and rolling, no protective atmospheres are used. Removing scale is applied.

 - In conclusion, it would be useful to compare the current technology of forging with CWR.

Response: The main purpose of the article is to determine the possibility of predicting cracks during cross - wedge rolling. The tests were carried out on the example of a harrow tooth preform. The shape of harrow tooth preform in the forging process is different and the process itself requires a greater number of operations. Therefore, it is difficult to compare anything.

 - In my opinion, the process of creation crack is most affected by the friction between the tool and the workpiece and by the temperature change in the formed material. These aspects of the process have not been specified by the authors.

Response: The process of creation crack does not depend on the friction between the tool and the billet but it depends on the initial temperature of billet. Cracks occur in the axial zone where the temperature is constant and the process takes place in isothermal conditions in the axial zone of billet. Therefore critical values of damage were determined at 1150 °C.

Reviewer 2 Report

in line 44, p.2, Rp is not mentioned. 

 In line 77, page 2 and line 86, p.3,  α and β would be described in Fig.1. It is helpful to understand.

In line 102, p.3, c.a 24mm, c.a. is not mentioned.

The sentence in line 128, p.5  is not completed. 

The reference is required in the material model in eq.(13) .

Author Response

Thank you for the review of the article and good comments.

Suggested changes have been included in the manuscript.The revised manuscript has been attached.
